# Inositol Hexaphosphate (IP6) and Colon Cancer: From Concepts and First Experiments to Clinical Application

**DOI:** 10.3390/molecules25245931

**Published:** 2020-12-15

**Authors:** Ivana Vucenik, Ana Druzijanic, Nikica Druzijanic

**Affiliations:** 1Department of Medical and Research Technology, University of Maryland School of Medicine, Baltimore, MD 21201, USA; 2Department of Pathology, School of Medicine, University of Maryland, Baltimore, MD 21201, USA; 3Department of Oral Medicine and Periodontology, School of Medicine, Dental Medicine, University of Split, 21000 Split, Croatia; ana.druzijanic8.1.ad@gmail.com; 4Department of Surgery, University Hospital Split, School of Medicine, University of Split, 21000 Split, Croatia; ndruzija@gmail.com

**Keywords:** cancer prevention, cancer treatment, phytic acid, molecular interactions, colon cancer, clinical observations

## Abstract

Multiple human health-beneficial effects have been related to highly phosphorylated inositol hexaphosphate (IP6). This naturally occurring carbohydrate and its parent compound, *myo*-inositol (Ins), are abundantly present in plants, particularly in certain high-fiber diets, but also in mammalian cells, where they regulate important cellular functions. However, the striking and broad-spectrum anticancer activity of IP6, consistently demonstrated in different experimental models, has been in a spotlight of the scientific community dealing with the nutrition and cancer during the last several decades. First experiments were performed in colon cancer 30 years ago. Since then, it has been shown that IP6 reduces cell proliferation, induces apoptosis and differentiation of malignant cells with reversion to normal phenotype, affecting several critical molecular targets. Enhanced immunity and antioxidant properties also contribute to the tumor cell destruction. Although Ins possesses a modest anticancer potential, the best anticancer results were obtained from the combination of IP6 + Ins. Here we review the first experimental steps in colon cancer, when concepts and hypotheses were put together almost without real knowledge and present clinical studies, that were initiated in colon cancer patients. Available as a dietary supplement, IP6 + Ins has been shown to enhance the anticancer effect of conventional chemotherapy, controls cancer metastases, and improves quality of life in cancer patients. Emerging clinical and still vast amount of experimental data suggest its role either as an adjuvant or as an “alternative” to current chemotherapy for cancer.

## 1. Introduction

Cancer is one of the major world’s health problems and challenge, and its incidence and mortality are still rapidly growing worldwide, despite enormous efforts to search for cure.

Colorectal cancer (CRC) is the third leading cause of cancer death in the world, and its incidence is steadily rising in developing nations [1]. CRC is the second most common cause of cancer death in the United States, and the third most common cause of cancer death in both men and women in the United States [2,3]. However, more than one-half of cancer cases and deaths are potentially preventable due to modifiable risk factors, such as smoking, unhealthy diet, consumption of alcohol, physical inactivity and overweight and obesity [4]. A simple modification of diet by increasing vegetable and fruit intake, by maintenance of optimum body weight and regular physical activity, it has been shown that 30–40% of cancers could be prevented [5,6]. The role of natural dietary agents in cancer prevention and treatment of both solid and hematologic malignancies has been recognized as a promising tool to control the onset and progression of malignancies, because of the little of no toxicity, high efficacy in multiple sites, oral consumption, known mechanisms of action, low cost and the human acceptance [7,8,9]. Since diet has an important role in the etiology of CRC [9], that is a very heterogenous disease caused by the interaction of genetic (5–10%) and environmental factors, the use of natural products for prevention and management of CRC is extremely promising.

IP6 (*myo*-inositol hexaphosphate, InsP_6_, phytic acid) and its parent compound *myo*-inositol (Ins) are abundant in plants, particularly in cereals and legumes [10,11,12]. The phosphate grouping in positions 1, 2 and 3 (axial-equatorial-axial) is exclusive for IP6, providing a specific interaction with iron to completely inhibit its ability to catalyze hydroxyl radical formation, making IP6 a strong antioxidant [13]. However, almost all mammalian cells contain IP6 and lower inositol phosphates, although in much smaller amounts, where they are important in regulating vital cellular functions and activities, such as cell division, cellular differentiation, ion channel permeability, embryonic development, but the most important role of these versatile inositol phosphates is their role in cellular transduction systems [14,15,16,17,18]. And this unique molecule, IP6, appears to be coming from the soil to our cells via the cereal grains, a staple for much of the world population.

Here we give a brief review of the anticancer potential of IP6, but providing a historical perspective, highlighting the first experimental steps utilizing IP6 against colon cancer, when concepts and hypotheses were put together almost without real knowledge, and also present the first clinical observational study in colon cancer patients, that was the base of several randomized controlled clinical trials that were conducted later.

## 2. Anticancer Activity of IP6

### 2.1. Hypotheses and First Experiments

Based on epidemiological data indicating that only diets containing a high IP6 content (cereals and legumes) showed a negative correlation with colon cancer, Shamsuddin et al. [19,20,21], performed pioneering experiments demonstrating novel anticancer feature of IP6 in colon cancer giving IP6 in a drinking water to experimental animals. He proposed that orally given IP6 would be absorbed through the gastrointestinal tract, that by the action of phytases and phosphatases in the diet and in intestinal cells, it would be quickly dephosphorylated to lower phosphorylated forms, and by entering the inositol phosphate pool, would affect tumor formation and progression. And that was a time when nobody believed that highly negatively charged IP6 could cross the intestinal barrier. He further proposed that the addition of Ins, would augment the anticancer activity of IP6 [19,20,21]. And, since inositol phosphates are common molecules involved in signal transduction in almost all mammalian cell systems, he further hypothesized that the anticancer action of inositol phosphates would be observed in different cells and tissue systems [19,20,21]. To summarize these early hypotheses —Hypothesis I: Because IP6 is common molecule in most mammalian cells, its anticancer actions will be observed in different cells and tissue systems; Hypothesis II: Addition of Ins to IP6 may enhance the anticancer function of IP6, because a higher amount of lower inositol phosphates (IPs) are expected to be formed; Hypothesis III: IP6 exerts its anticancer functions through lower IPs by entering into IP intracellular pool. And indeed, all of these proposed hypotheses have been tested over the years and confirmed in subsequent experiments.

In testing the hypothesis that the anticancer action of IP6 may be mediated via lower phosphorylated forms of inositol, we performed experiments and showed that IP6 was not only absorbed from the gastrointestinal tract, but also taken up by malignant cells [22]. In fact, orally administered IP6 was able to reach target tumor tissue distant from the gastrointestinal tract [23]. Exogenous IP6 was rapidly taken up by mechanisms involving pinocytosis or receptor-mediated endocytosis, transported intracellularly, and dephosphorylated into inositol phosphates with fewer phosphate groups [22]. To confirm the hypothesis and the proposed mechanism of action, we followed the fate of radiolabeled IP6. Analyzing absorption, intracellular distribution, and metabolism of [^3^H]-IP6 in tumor cells by differential centrifugation and anion-exchange chromatography, we found that the predominant metabolite of IP6 was inositol tetraphosphate (IP4), indicating the possible important role of this lower IP6 metabolite in its anticancer activity [22]. Utilizing the same approach, and analyzing the uptake, distribution, and metabolism of [^3^H]-IP6 in tissues, plasma, urine and feces, after it was given orally to rats bearing mammary tumors, we found that it was quickly absorbed from the gastrointestinal tract and widely distributed through the body, with a substantial amount of radioactivity found in tumor tissues [23]. These data showing a rapid distribution and metabolism of exogenously given IP6, at least in part explained the anticancer activity of IP6 at sites distant from the gastrointestinal tract. However, determination of non-radiolabeled IP6 in biological matrices, fluids and tissues has always been a big problem, obstacle, and challenge. Appropriate analytical methodologies were developed and utilized by Prof. Grases and his group in Spain [24,25]. Using inductively coupled plasma atomic emission spectrometry (ICP-AES), and later liquid chromatography-mass spectrometry (LC-MS) bioanalytical methods, they were able to develop direct, sensitive, and selective bioanalytical methods for determination of IP6 and other inositol phosphates in mammalian cells and tissues. A recently introduced the polyacrylamide gel electrophoresis (PAGE)-based method is very suitable for detection of IP6 and other inositol phosphates not only in plants, but also in animal and human cells and tissues [26]. Ferry et al. [27] added further confirmation to these data and the original hypothesis, as well as the pioneering observation that the externally applied IP6 enters the cell followed by dephosphorylation. Eiseman et al. [28] also observed an increase in dephosphorylated metabolites of IP6 tumor xenografts in IP6-fed mice. Furthermore, that highly negatively charged IP6 molecule is internalized by endocytosis as a cation salt and revealed that, subsequently, IP6 is completely dephosphorylated to inositol in lysosomes [29]. Thus, not only one of the key misconceptions about this molecule is nullified, but it has been demonstrated that IP6 is an essential nutrient whose level in plasma and urine fluctuates following deficiency or replenishment [24,25]. In essence, it has all the characteristics of a vitamin. Interestingly, very recently, while critically discussing employing IP6 as a new options of cancer treatment, a group from Germany postulated and confirmed almost the same hypothesis [30].

### 2.2. IP6 Is a Broad-Spectrum Anticancer Agent

The vast amount of laboratory data accumulated over 30 years in vitro and in vivo has confirmed the original Hypothesis I, that IP6 is indeed a broad-spectrum antineoplastic agent, effective against cancers of different cells and different tissue systems.

The most important hallmark of cancer is a sustained growth and a sustained chronic proliferation. A consistent growth inhibition of many different cancer cells has been shown for IP6 in a time- and dose-dependent fashion, and in concentrations from 0.5–5.0 mM. Antiproliferative effects of exogenous and extracellular IP6 were observed on human leukemia cells [31,32], human colon cancer cells [33] human breast cancer cell line, both estrogen receptor-positive and estrogen receptor-negative [34], cervical cancer [27], prostate cancer [35,36], and hepatoma cell lines [37]. In addition to all these epithelial cells, IP6 was able to inhibit the growth of mesenchymal tumors, such as murine fibrosarcoma [38] and human rhabdomyosarcoma [39]. However, different sensitivity to IP6 were observed with different cell types and in different models; leukemia cells and hepatocellular carcinoma cells responded more sensitive to IP6 treatment, indicating that this growth inhibitory effect was attributed to different mechanisms [40].

Along with this reduction in cell proliferation, rather normalization, IP6 induces differentiation and maturation of malignant cells, often resulting in reversion to the normal phenotype, as demonstrated in K-562 hematopoietic cells [31], human colon carcinoma HT-29 cells [33,41], prostate cancer cells [35], breast cancer cells [34] and rhabdomyosarcoma cells [39]. It was also observed that IP6 treatment of human hepatoma cell line caused a dramatic reduction in secretion of the tumor marker α-fetoprotein (AFP) [37]. A practical application of this effect in the clinical setting is to monitor patients with cancer for reduction of tumor markers by laboratory tests, and there have been emerging evidence of that happening clinically.

The cancer preventive activity of IP6 in vitro was tested in a benzo[*a*]pyrene-induced transformation of rat tracheal epithelial cells [42], and then in a model using BALB/c mouse 3T3 fibroblasts [43] with modest efficacy. Further, IP6 was shown to be effective in prevention of skin cancer, utilizing in vitro model of skin carcinogenesis in JB6 cells [44] and in vivo model, where IP6 inhibited 7,12-dimethylbenz[*a*]anthracene (DMBA)-induced mouse skin tumor development, indicating its potential for the management of skin tumorigenesis [45].

As stated before, the very first experiments with IP6 were investigating the effectiveness of IP6 as a cancer preventive agent in colon cancer and showed a great potential in models where cancer was induced in different species (rats and mice) and even with different carcinogens (1,2-dimethylhydrazine and azoxymethane) [19,20,21,46,47,48,49,50,51]. It is important to highlight that IP6 was effective in preventing tumor formation either given before or after carcinogen administration. However, extremely significant was a finding that IP6 was potent in reducing development of large intestinal cancer, even when treatment was begun 5 months after carcinogen initiation. Compared to untreated rats, animals on IP6 had 27% fewer tumors. These findings pointed towards its possible therapeutic use [21]. Aberrant crypts, putative preneoplastic lesions, have been proposed as intermediate biomarkers for colon cancer, and in these first studies of colon cancer, IP6 was shown to be effective in decreasing the incident of aberrant crypt foci in IP6-treated rats [48,49]. Studies were expanded to other experimental models. A consistent, reproducible, and significant inhibition of mammary cancer by IP6 was shown in experimental models chemically induced either by DMBA [52,53,54,55] or *N*-methylnitrosourea (NMU) [47] by us, and others, and the effect was seen on tumor incidence, tumor size, and tumor multiplicity. Anticancer effect of IP6 was also shown in murine transplanted and metastatic fibrosarcoma [38] and in various models of skin cancer [45,56,57,58]. Protective effect of IP6 against ultraviolet B (UVB) light, known as a complete carcinogen, was shown in SKH1 hairless mice [57,58].

The therapeutic properties of IP6 were first shown in the FSA-1 mouse model of transplantable and metastatic fibrosarcoma, where after subcutaneous inoculation of mouse fibrosarcoma FSA-1 cells, mice were treated with intraperitoneal injections of IP6 [38]. IP6 treatment resulted in increased survival and a significant inhibition of tumor size. The same model was used to monitor the development of lung metastasis after the intravenous injections of FSA-1 cells, and IP6 treatment significantly reduced the development of lung metastases. In two other models, we monitored the effects of IP6 on tumor formation, and then on tumor regression. Effect of IP6 on tumor formation was tested in human rhabdomyosarcoma RD cells transplanted in nude mice. The peritumoral treatment with IP6 (40 mg/kg) started two days after subcutaneous injection of rhabdomyosarcoma cells, three times weekly for 5 weeks. There was a 49-fold reduction of tumor size in IP6-treated animals [39]. To test the efficacy of IP6 on tumor regression, we used similar model for liver cancer, trying to see if IP6 would be potent in inhibiting experimental hepatoma [37,59]. First, a single treatment of HepG2 cells in vitro by IP6 resulted in the complete loss of the ability of these cells to form tumors when inoculated subcutaneously in nude mice [59]. Additionally, we demonstrated that the preexisting liver cancers regressed when treated directly with IP6 [59]. The treatment with intra-tumoral injection of IP6 started when tumor reached 1 cm in diameter, and after 12 days of treatment, tumor weight in IP6-treated mice was 3.4-fold less than in control mice [59]. Also, IP6 administered in drinking water was a potent inhibitor of in vivo growth of human prostate cancer xenografts in nude mice [60].

### 2.3. Ins Potentiates the Anticancer Activity of IP6

Ins itself was also demonstrated to have anticancer function. It inhibited colon, mammary and lung tumor formation [20,52,61,62]. Additionally, as hypothesized (Hypothesis II), it has been shown that Ins potentiates both the antiproliferative and antineoplastic effects of IP6 in vivo [20,38,52]. Synergistic cancer inhibition by IP6 when combined with inositol was observed in colon cancer (Table 1) [20].

Similar results were seen in the mammary cancer studies [52] and in metastatic lung cancer model [38]. Of note is that in some models some tumor parameters may either remain unchanged or worse, exacerbated by Ins or IP6 alone. However, not only that the combination of IP6 and Ins was significantly better in different cancers than was either one alone, but it also consistently reduced all the tumor parameters. This initial hypothesis that Ins potentiates the anticancer effect of IP6, has been recently confirmed by a series of studies conducted by Dr. Song Yang and his group investigating colon cancer and its metastasis to the liver [64,65,66] showing that when combining IP6 and Ins the survival was improved, while the tumor mass and liver metastasis were decreased. These findings are of a special importance because metastasis is a primary cause of death in colon cancer patients, with a liver as the most common site [64,65,66].

Thus, in clinical settings, one must not use IP6 or Ins alone since individually they are neither optimally efficient, nor sufficient; only together, they work the best.

### 2.4. Mechanism of Action and Molecular Interactions of IP6

Though we know that the cellular mechanisms involved in the anticancer activity of IP6 and Ins involve cell proliferation and differentiation, we now better understand the involvement of these molecules in biochemical pathways and their molecular interaction. It is known that virtually all animal cells contain inositol phosphates that have important role in cellular signal transduction, regulation of cell function, growth, and differentiation [14,15,16,17,18]. The role of IP6 among multiple signaling pathways and their cross-talks in regulation of cell function is complex and still needs to be addressed in the future.

Detailed and elaborate reviews of the anticancer activity of IP6 and Ins have been published [30,40,67,68,69]. Here we highlight some of the most important cellular mechanisms and molecular targets of IP6 and focus on those important in colon cancer research.

IP6 can modulate cellular response at the level of receptor binding; after sterically blocking the heparin-binding domain of basic fibroblast growth factor, IP6 disrupted further receptor interactions [70]. In addition to acting on phosphatidylinositol-3 kinase and activating protein-1 by IP6 [44], protein kinase C [71,72] and mitogen-activated protein kinases [44,73] are involved in IP6-mediated anticancer activity. Various cellular proteins, pathways and mechanisms are involved in the observed anticancer effects of IP6. In addition to the growth inhibition, induction of differentiation and apoptosis, inhibition of angiogenesis and suppression of tumor metastasis are observed [30,40,67,68,69]. Critical molecular targets are involved in these cellular processes and activities. IP6 downregulates p21 and p27, inhibits pRB phosphorylation and cell cycle progression [72], targets PI3K/Akt pathway and counteracts the activation of PKC/RAS/ERK pathway [30,67,68,72]. Furthermore, by downregulating Akt and ERK, IP6 reduces NF-kappaB and inhibits inflammation [30,67,68]. In summary, IP6 impacts various intracellular signaling pathways and networks, and expression of genes encoding key cellular proteins, such as p53, p21, p27, BCL-2, and MMPs. It seems that the inhibition of the phosphorylation-based activation of key molecular targets is a basic mechanism through which IP6 interferes with specific cellular and biological functions [67]. Very recently, the regulation of Cullin-RING E3 ligase dynamics by IP6 has been suggested [74]. Most of the current therapeutic molecular glues that target disease-causing proteins for degradation utilize the largest family of ubiquitin ligases in humans, the Cullin-RING (Really Interesting New Gene) ligases (CRLs). Because, IP6 causes G1 arrest by down-regulating p21 and p27, and since both p21 and p27 are well-defined substrates for CRL1- and CRL4-based ubiquitin ligases, it has been tempting to speculate that at least some degree of IP6’s anticancer activity might come from increased down-regulation of CRL function, thus stabilizing p21 and p27 to promote G1 arrest [74].

Investigating the anticancer effect of IP6 in colon cancer, the involvement of already mentioned intracellular signaling pathways and critical targets have been shown in a broad-spectrum activities from the effect on aberrant crypt foci, an early biomarker of colon carcinogenesis [30,40] to the regulation of microRNA-155 and its related genes expression [75]. Exploring the effect of combined IP6 and Ins on the appearance of colorectal cancer metastasis to the liver, Dr. Song and his group conducted several studies. Here, utilizing a splenic capsule, first they were trying to see the effect, and after to explore and understand the mechanism of this metastatic spread [64]. Tumor metastasis is a complex process, involving adhesion, migration, intra- and extravasation, colonization, and proliferation, with alterations of multiple mechanisms. Major components of the cell-extracellular interactions are collagen IV, fibronectin, and laminin, and IP6 and Ins treatment altered the expression of all these proteins. Additionally, MMP-9, a type IV collagenase, and its high expression is closely correlated with malignant tumor invasion, metastasis and vascular formation, and IP6 and Ins were shown to inhibit the MMP-9 in this model [64]. Very recently, in the follow-up study, the same group investigated the liver metastasis of colorectal cancer, this time using an orthotopic transplantation model of colorectal cancer to monitor the effect of IP6, Ins and their combination. First, they show again, that the IP6 + Ins was more effective in inhibiting the liver metastasis of colon cancer than each compound administered individually. Additionally, regulation of mutation of Wnt/β-catenin signaling pathway by inhibiting Wnt10b, Tcf7, β-catenin, and c-Myc from abnormally high expression is critical to increase efficacy [65].

Several other mechanisms can also be involved. First, antioxidant function of IP6 is best known and recognized [13,76]. Through its unique structure, IP6 can chelate Fe^3+^ and suppress the hydroxyl radical formation. Consequently, IP6 can inhibit the carcinogenic process mediated by active oxygen species and prevent cell injury by its antioxidative function. Mineral binding activity of IP6 can further contribute to its anticancer activity; binding with Zn^2+^ IP6 can affect activity of thymidine kinase, an enzyme that is essential for DNA synthesis. Additionally, IP6 can remove the excess iron, which may facilitate and augment colorectal cancer IP6 [46,51].

Anticancer effect of IP6 is not only focused on tumor cells, but also on a host. IP6 is known for immune support, being able to restore its immune system. IP6 was shown to augment the activity of natural killer (NK) cells in vitro, and in vivo it was shown to be able to normalize the carcinogen-induced depressed NK activity [63,77]. As shown in Table 1, there is an inverse relationship between tumor incidence and NK-cell activity in a mouse model of colon cancer. IP6-induced enhancement of NK-cell activity correlates with suppression [63]. Although, both IP6 and Ins show lower incidence of cancer and enhanced NK-cell activity, animals that received IP6 + Ins combination had the lowest cancer incidence with the highest NK-cell activity [63].

### 2.5. Sensitivity and Selectivity of IP6

Almost all the anticancer agents have severe side effects on normal tissues and organs. IP6 was shown that while it can sensitize cancer cells for treatment, at the same time can offer a protection to normal tissues. This sensitivity and selectivity of IP6 is its unique property. In one of the first studies in Italy, using bone marrow and isolated fresh CD34^+^ cells from bone marrow and observing the treatment with different dose of IP6. While IP6 was very effective against the leukemic progenitors from chronic myelogenous leukemia patients, no cytotoxic or cytostatic effect was observed on normal bone marrow progenitor cells under the same conditions [32]. Investigating the effect of IP6 on Kaposi Sarcoma, IP6 inhibited the colony formation of Kaposi Sarcoma cell lines, KS Y-1 (AIDS-related KS cell line) and KS SLK (Iatrogenic KS), and CCRF-CEM (human adult T lymphoma) cells in a dose-dependent manner [78]. However, IP6 did not affect the ability of normal cells (peripheral blood mononuclear cells and T cell colony-forming cells) to form colonies in a semisolid methylcellulose medium [78]. The activity of IP6 was compared to taxol, that was used as a control, and that contrary to IP6 affected the colony formation of normal cells. A similar effect was observed when the response of MCF7-10A cells, a normal human breast epithelial cells was compared with the breast cancer cell lines MCF-7 and MDA-MB-231. While the growth of the epithelial cells was not affected, the proliferation of cancer cells was significantly reduced after the incubation with IP6 due to the increased rate of apoptosis [79]. There are known differences between malignant and normal cells in their behavior, known as the hallmarks of cancer, and described by Hannahan and Weinberg, and differences in metabolism, known as Warburg effect. However, this selectivity of IP6 for normal and malignant cells deserves further investigation.

Furthermore, IP6 was shown to acts synergistically with standard chemotherapeutics. Cancer therapy recognizes the importance of using combination therapy with rationale to increase efficacy and decrease side-effects of conventional chemotherapy. Another important aspect of combination therapy is overcoming acquired drug resistance. It was demonstrated that IP6 acted synergistically with doxorubicin and tamoxifen, being particularly effective against estrogen receptor-negative and doxorubicin-resistant cell lines, both conditions that are very challenging to treat [80]. These data are particularly important because tamoxifen is usually given as a chemopreventive agent in the post-treatment period, but also because doxorubicin has enormous cardiotoxicity and its use is associated with doxorubicin resistance, both conditions improved by IP6.

## 3. With IP6 and Ins in Clinics Today

Our first clinical observational study was on colon cancer patients given IP6 + Ins during chemotherapy; no toxicity, no drop in blood cell count, no tumor progression and improved quality of life were observed. Available as dietary supplements, both with a GRAS status, IP6 and Ins have been in clinical practice for almost twenty years. During that time, many case reports, anecdotal evidence, and few small clinical studies have indicated and demonstrated an enhanced antitumor activity with improved quality of life by IP6, Ins and their combination. In today’s era of the evidence-based medicine, a practice-based evidence is needed. Therefore, case reports with detailed information add to our medical knowledge and are valuable in our clinical care. Overall, in all these reports, a reduced tumor growth rate was noticed with IP6 + Ins, and even in some cases a regression of primary lesions. Moreover, when IP6 + Ins was given in combination with chemotherapy, side effects of chemotherapy, such as drop in leukocyte and platelet counts, nausea, vomiting, alopecia, were diminished and patients were able to perform their daily activities, shown mostly in breast and colon cancer [81,82]. Although majority of case studies deal with breast and colon cancer, there was a report on a patient with advanced non-small cell lung cancer in Japan. A patient with a smoking history of 30 years, was treated by chemo-radiotherapy (CRT) with partial response. Eight months later, following the CRT, she began to take IP6 + Ins and almost 5 years later she was enjoying a completely healthy life without any signs of relapse [83]. In a phase I clinical trial Ins was shown to be safe and well-tolerated [84], and this study was extended to randomized, double blind, placebo-controlled, phase IIb study to determine the chemopreventive effects of myo-inositol in smokers with bronchial dysplasia. In the treatment of patients with advanced malignancy receiving chemotherapy, the combination of beta-(1,3)/(1,6) D-glucan and IP6 had beneficial effects, particularly on hematopoiesis [85]. Small, prospective, randomized, pilot clinical study, was conducted in Croatia showing that IP6 + Ins decreased the negative side effects of chemotherapy and improved quality of life in breast cancer patients [81]. In a double-blind, randomized controlled trial (RCT), even when applied topically, IP6 treatment was effective and safe in preventing chemotherapy-induced side effects as well as maintaining quality of life in women with ductal breast cancer [86]. When the literature search was conducted with a goal to define and show clinical evidence of the effectiveness of IP6 and Ins on the quality of life in cancer patient with breast cancer, it showed and confirmed that IP6 and Ins were effective in improving quality of life of patients undergoing chemotherapy [87]. Khurana et al. published an amazing case report on a metastatic melanoma [88]. A patient with metastatic melanoma declined traditional therapy and opted to try the IP6 + Ins supplement only. He received a complete remission and remained in remission 3 years later. Because immunotherapy is a treatment option for melanoma, and immunomodulatory function of IP6 has been shown, in addition to its selectivity, this could be a new avenue for IP6 in clinical practice.

## 4. Conclusions and Future Directions

Since currently available preclinical and encouraging clinical data suggest that IP6 and Ins are promising in cancer prevention and adjuvant therapy, more controlled clinical trials are needed and expected. Although the decrease in overall cancer frequency and death rates is encouraging, the rising incidence and mortality for some cancers are of concern; therefore, the potential therapeutic benefits of IP6 and Ins have become increasingly relevant. Colorectal cancer (CRC) is on the rise in USA among young people. Currently, CRC is the fourth most common cancer in the United States. The National Cancer Institute estimates that in 2020 there will be 104,610 new cases of colon cancer and 43,340 cases of rectal cancer. There will be about 53,200 deaths from CRC. And IP6 + Ins, as proved by experimental and clinical data, can be a new option not only for cancer prevention, but also for cancer treatment.

However, a recent report that IP6 can increase platelet aggregate size [89] needs to be addressed. Additionally, it appears that platelets have ability to interact with tumor cells in the circulation, contributing to the “circulome” formation [90]. Cancer-associated thrombosis has been a conundrum in managing cancer patients for a long time because persons with cancer are four to seven times more likely to develop thrombosis than the general populations [91]. On the other hand, a strong in vitro anticoagulant activity of Na-IP6 has been demonstrated in the blood of various animals, showing also that sodium phytate formulations were able to inhibit platelet aggregation [92]. The sodium salt of phytic acid (phytin) showed strong anticoagulant activity in vitro and further studying the effect of phytin on coagulation time, it was observed that phytin possessed significant anticoagulant activity in vivo as well [93]. Investigating the effect of IP6 on platelet aggregation, our own data show that agonist-induced platelet aggregation was markedly inhibited by IP6 in rats given IP6 in drinking water [94], and in healthy volunteers [95]. tThese new findings, new data and new theories need to be addressed in vivo and in clinical practice, monitoring the physiological and biological responses.

## Figures and Tables

**Table 1 molecules-25-05931-t001:** Cancer inhibition by IP6 when combined with Ins in 1,2-dimethylhydrazine (DMH)-induced colon carcinoma in mice and on Natural Killer (NK)-cell activity.

Experimental Group	Tumor Incidence (%)	Total Number of Tumors	Number of Tumors/Tumor Bearing Mice	Mitotic Rate (%)	NK-Cell Activity (%)
DMH	63 *	22	12	1.92 ± 0.17	19.4 ^+^
DMH + IP6	47 **	13	10	1.48 ± 0.15	31.7 ^++^
DMH + Ins	30	9	6	1.01 ± 0.14	32.1
DMH + IP6 + Ins	25 ***	4	4	1.06 ± 0.13	39.8 ^+++^

Note that animals on IP6 + Ins fared the best either in appearance of microscopic and macroscopic carcinomas or in NK-cell activity. The difference in tumor incidence between DMH * and DMH + IP6 + Ins *** is significant at *p* < 0.001, and between DMH + IP6 ** and DMH + IP6 + Ins *** at *p* < 0.005. For other parameters, the statistical difference is the same. The enhancement of NK-cell activity is inversely correlated with tumor incidence. The difference in NK-cell activity between DMH ^+^ and DMH + IP6 ^++^ is significant at *p* < 0.01 and between DMH ^+^ and DMH + IP6 + Ins ^+++^ is significant at *p* < 0.005. Adapted from Shamsuddin et al. 1989 [20] and Baten et al. 1989 [63] for NK-cell activity.

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
