# Peer review of "Inositol Hexaphosphate (IP6) and Colon Cancer: From Concepts and First Experiments to Clinical Application"

_molecules, 2020, doi:10.3390/molecules25245931_

Round 1
Reviewer 1 Report
Excellent and comprehensive review on polyphosphate inositol from experimental evidence to clinical application. It is well written and worth to be published.
Author Response
Reviewer 1
Comments and Suggestions for Authors
Excellent and comprehensive review on polyphosphate inositol from experimental evidence to clinical application. It is well written and worth to be published.
Answer to Reviewer 1
Thank you for these wonderful comments.
The revised paper has been checked for the style and accuracy.
Reviewer 2 Report
molecules-1004775
General comments
This manuscript provides the information of phytic acid and its relevant molecules for cellular physiology and molecular pathology. In addition, it also describes unpublished data of preliminary clinical study of the authors. However, this manuscript includes several flaws, as described below.
Major concerns
- IP6 may be an uncommon abbreviation for readers. Therefore, in the title, IP6 should be spell out. In abstract, the synonym of IP6, like line 54, should be included.
- Please expand IP4, ICP-AES, LC-MS for the first-time use, like PAGE.
- The sentences Line 95-110 and 127-141 seems to be duplicated. Please also carefully check any duplicated texts in the manuscript.
- In main text, the literature should appear in numeral order.
- Line 194, myo-inositol (Ins) has already appeared in Line 54.
- Table 1. Please explain what NK-cell activity means.
- Table 1. Were there any statistical differences, except for the tumor incidence?
- Line 268. The word “?OH” is possibly garbled.
- The section 3 is an unpublished original article, not a literature review. In addition, it seems to be out of date and probably out of current standard medicine. It is a controversial issue that whether these data should be included in the manuscript. Please declare it in abstract if the authors include the section 3.
Minor Points
- This manuscript includes inappropriate spaces and several fonts irregularly. Please carefully check them.
- GRAS (Line 315) and GRAAS (Line 388). Which is correct?
Author Response
Answer to Reviewer 2
Dear reviewer,
Thank you for your very constructive comments that scientifically and aesthetically greatly improved this manuscript
General comments
This manuscript provides the information of phytic acid and its relevant molecules for cellular physiology and molecular pathology. In addition, it also describes unpublished data of preliminary clinical study of the authors. However, this manuscript includes several flaws, as described below.
Answer
Here, we wanted to give an overview of the anticancer activity of inositol hexaphosphate (IP6). Thank you again for your efforts to improve the quality of this manuscript.
Major concerns
- IP6 may be an uncommon abbreviation for readers. Therefore, in the title, IP6 should be spell out. In abstract, the synonym of IP6, like line 54, should be included.
- Please expand IP4, ICP-AES, LC-MS for the first-time use, like PAGE.
- The sentences Line 95-110 and 127-141 seems to be duplicated. Please also carefully check any duplicated texts in the manuscript.
- In main text, the literature should appear in numeral order.
- Line 194, myo-inositol (Ins) has already appeared in Line 54.
- Table 1. Please explain what NK-cell activity means.
- Table 1. Were there any statistical differences, except for the tumor incidence?
- Line 268. The word “?OH” is possibly garbled.
- The section 3 is an unpublished original article, not a literature review. In addition, it seems to be out of date and probably out of current standard medicine. It is a controversial issue that whether these data should be included in the manuscript. Please declare it in abstract if the authors include the section 3.
Answers
- IP6 has been spelled out in the title as suggested.
- IP4, ICP-AES, LC-MS has been spelled out, as is custom when the abbreviation is used for the first time. Sorry for this omission. IP4 (inositol tetraphosphate) (line 99), ICP-AES (inductively coupled plasma atomic emission spectrometry) (line 109), LC-MS (liquid chromatography-mass spectrometry) (line 110).
- The sentences in the Lines 95-127-141 and 127-141 have been checked. Yes, that was the same paragraph, and it was my mistake: a big apology. A new paragraph is added. Additionally, the whole manuscript was carefully checked for any possible duplicated text.
- Because of the mistake #3 and using the same paragraph, the mistake happened. Adding a real-new paragraph (lines 130-140), the references came back in order. While writing we were trying to keep them in numerical order. They are carefully checked again to appear numerically. However, some references were mentioned before, and seemed to be out of order. Again, a big apology.
- Line 194, myo-inositol was abbreviated, as suggested. Ins (line 193).
- Table 1. NK-activity was spelled out. And, it was checked again that this natural killer-cell activity was explained in the text (line 199).
- Other statistical analysis and differences for other parameters are added in the table for the tumor appearances and for the NK-cell activity (lines 200-206). NK-cell activity is added in the text (lines 281-284).
- Line 268 “?OH”…… The unpaired electron of the hydroxyl radicalis officially represented by a middle dot. It is ·OH…to avoid any future confusions, the symbol is replaced with the word “hydroxyl radical” (line 272).
- The subtitles in the section 3 are changed, so that it will not look like “unpublished the original article”: 3.1. Material and Methods (line 322) and 3.2. Results and Comments (line 338) are added, while 3.3 Discussion (line 374) is deleted. These data are basic laboratory tests used for monitoring of cancer patients, and the same parameters have been used in clinical laboratories for several decades. The difference might be in expression of results, conventional or international units. However, more importantly is a consistency in monitoring; the results were obtained from the same laboratory, using the same approach and the same methodology, so that the results could be compared. The testing was addressed (lines 345-347). These data are important to be included to show that IP6 + Ins did not influence the results (no toxicity). This study was included in the Abstract, please see lines 29-30. However, a mistake was made in expressing Erythrocytes in Table 3. It needs to be Erythrocytes (E) (x 1012/L), not as listed Erythrocytes (E) (x 109/L).
Minor points
- This manuscript includes inappropriate spaces and several fonts irregularly. Please carefully check them.
- GRAS (Line 315) and GRAAS (Line 388). Which is correct?
Answers
1.The document is carefully checked for inappropriate spaces and font irregularity. Thank you for this note.
- GRAS is correct (line 318). It is corrected in line 392.
Round 2
Reviewer 2 Report
The authors properly replied the comments of the reviewers, and the revised manuscript has been considerably improved and suitable for the publication.